# Investigation of the Conductive Properties of ZnO Thin Films Using Liquid Probes and Creation of a Setup Using Liquid Probes EGaIn for Studing the Conductive Properties of Thin Films

Nikita Permiakov , Evgeniya Maraeva *, Anton Bobkov, Ritsoh Mbwahnche and Vyacheslav Moshnikov

Department of Micro-and Nanoelectronics, Faculty of Electronics, Saint-Petersburg Electrotechnical University "LETI", 5, pr. Popova, Saint-Petersburg 197022, Russia
* Correspondence: evmaraeva@etu.ru; Tel.: +7-952-212-5052

**Abstract:** The use of liquid probes based on indium–gallium eutectic (EGaIn) with the possibility of positioning is an important problem for the study of thin films. This work is centered on the creation of a setup for measuring the current–voltage characteristics with the use of a liquid eutectic electrode. A technique for obtaining a cone-shaped liquid EGaIn electrode, a 3D assembly model and an algorithm for the operation of a probe setup for obtaining the current–voltage characteristics using liquid contacts are presented.

**Keywords:** current–voltage characteristic; indium–gallium eutectic; liquid probe; zinc oxide



## 1. Introduction

Currently, the use of liquid probes based on EGaIn with the possibility of positioning is an important application for the study of film samples, such as conductive organic films, the study of the effect of the meniscus shape on recorded signals, and the study of materials for flexible electronics [1]. At the nanoscale, nanodroplets of liquid metals simultaneously possess a high electrical and thermal conductivity, tunable reactivity and useful physicochemical properties. They also offer special fusion and alloy removal conditions for the formation of liquid-based multi-element nanoalloys or the synthesis of engineered solid nanomaterials [2–4].

In [5], studies of the drop shape of indium–gallium eutectic melt were carried out. For a needle tip with a diameter of 400 μm, it was possible to obtain an electrode tip with a diameter from 50 to 100 μm. Even after the needle was sharply retracted from the drop, the eutectic drop did not turn into a hemisphere. The contact pressure of the liquid probe on the test surface decreased with an increase in the diameter of the contact area. Liquid eutectic melt in air is covered with a thin layer of gallium oxide. This oxide layer acts as a membrane, the surface tension of which determines the shape of the droplet. Upon reaching a small critical surface tension, a thin layer of oxide breaks and immediately reforms as a result of the interaction with air through microfluidic channels. It turns out that a drop of eutectic indium–gallium melt constantly has an oxide membrane on its surface.

Thus, the oxide film is an excellent surfactant for metals, and the liquid metal can be removed and transformed quickly and reversibly [6].

On the surface of the eutectic indium–gallium melt there is a layer of gallium oxide, which is an n-type semiconductor. The melting temperature of the melt is 15.5 °C, the conductivity is $3.4 \times 10^4$ S/cm and the work function of EGaIn is 4.1–4.2 eV [7]. Based on the fact that it is a good conductor, in [8], it was proposed to use 3D structures of liquid-phase GaIn alloy embedded in an elastomer. This allows the production of flexible 3D structures that do not lose their conductive properties when deformed. EGaIn, which is introduced into microchannel, is a liquid at room temperature, and is able to withstand significant

mechanical deformations without destruction or loss of electroconductive properties, at the same time, since the melt is in the capillaries, it will not spread.

It should be noted that important features of liquid contacts are the possibility of transmitting high currents, the formation of an electrical breakdown and the creation of fractal structures [9–11].

There are various options for depositing planar circuits based on EGaIn: pouring into formed channels [12] and depositing through a stencil followed by cooling and sealing with an elastomer [13]. For three-dimensional schemes, the casting method is used, followed by freezing of the liquid eutectic EGaIn, or by the vacuum casting method. Following these operations, all structures are in the solid state at temperatures below room temperature. However, since they will operate at room temperature, the stacked chains must be filled with elastomer to prevent the spreading of the EGaIn structures as the temperature rises to room temperature.

Furthermore, liquid eutectic indium–gallium melt is actively used in microcontact printing [13]. This method of forming flexible conducting channels is widely used in soft lithography [14]. Despite the fact that this method is much slower than inkjet electrohydro-dynamic deposition, it is cheaper and can be carried out with liquid EGaIn directly in air by guiding the probe along a predetermined pattern. Microcontact printing is relatively easy using inks that easily wet the elastomer surface. The indium–gallium melt does not have this property due to the oxidation of its surface, as a result of which a large drop is formed and the surface tension forces are not easy to break [15]. However, this melt has the advantage of retaining the shape of the deposited structure during external piping and sealing. When the hemispherical tip is removed from the surface, a portion of the liquid alloy is transferred to the flat substrate and forms a drop. Droplets that are deposited at a distance smaller than the droplet diameter merge and form a continuous line. Once the pattern is printed, external leads are connected to it, the liquid melt freezes and is coated with an additional layer of elastomer. Individual drops with a diameter of 340 μm can be applied in increments of up to 13 μm. However, such a small step is not necessary, and already at a step of 200 microns, individual drops merge into a single line [13].

In [16,17], a four-probe method with liquid probes is used to study the carrier mo-bility of organic films. In this study [17], four drops of GaIn were collected using a four-needle scanning tunneling microscope to inject them into an independently controlled four-probe system.

Another interesting example of the use of EGaIn to create electrodes is shown in [18], where two types of electrodes were used—tungsten needle probes and GaIn electrodes—to monitor the features of the heart rhythm of insects. The fact is that obtaining an electro-cardiogram (ECG) of insects is difficult due to weak signals and a limited contact area for applying electrodes. Using an electromagnetically shielded heart signal acquisition system, consisting of analog amplification and digital filtering, ECG signals of three phenotypes with different heart functions were obtained. It has been shown that during long-term ECG recording, the non-invasive method implemented with GaIn electrodes is relatively stable, both in amplitude and in period.

Currently, the use of liquid probes makes it possible to deal with such questions that arise during testing, such as: how much pressure the probe puts on the sample, whether it can damage the test layer and how the current will flow after that. Nevertheless, there are still unresolved questions about what factors affect the contact resistance, or how many times the EGaIn electrode can be reused and how the contact area affects the measurement results [19].

The purpose of this work is to create a setup for measuring the voltage–current characteristics with the use of liquid eutectic electrode. Unfortunately, previous works related to the topic do not present specific algorithms for obtaining liquid probes and measurement algorithms. In this paper, Section 2.1 proposes a specific mechanism for obtaining a cone-shaped electrode; in Section 2—algorithm of operation with one elec-

trode, in Section 3—specific examples of obtained current–voltage characteristics of the ZnO samples.

## 2. Materials and Methods

Zinc oxide test samples were obtained as follows: ZnO layer was deposited on conductive FTO substrates using ultrasonic spray pyrolysis [20–22]; 0.1 M aqueous solution of zinc acetate served as the sprayed solution, the substrate temperature was 270 °C and the spraying time was 10 min. Subsequently, ZnO nanoparticles were grown on the substrates by the hydrothermal method. To do this, a substrate with a ZnO film applied by spray pyrolysis was lowered into an aqueous solution of zinc acetate, hexamethylenetetramine and cetyltrimethylammonium bromide, and kept in a thermostat at a temperature of 85 °C for 60 min. Once the given time had elapsed, the substrate was washed in distilled water and dried in the atmosphere at room temperature.

To implement the probe setup, a module was created using one insulin syringe with a needle outlet with a diameter of 400 μm. This module was designed and its parts were manufactured by 3D extrusion printing with ABS plastic [4]. For precise control, a KYSAN 1124090 stepper motor and a screw-nut movement mechanism were used. The thread pitch of the M5 screw is 0.8 mm. The stepper motor used has an angular step of 1.8°. Thus, in one step of the stepper motor, the moving part will move 4 μm.

### 2.1. Obtaining a Cone-Shaped Electrode

In Figure 1, the stages of obtaining a cone-shaped electrode are presented.

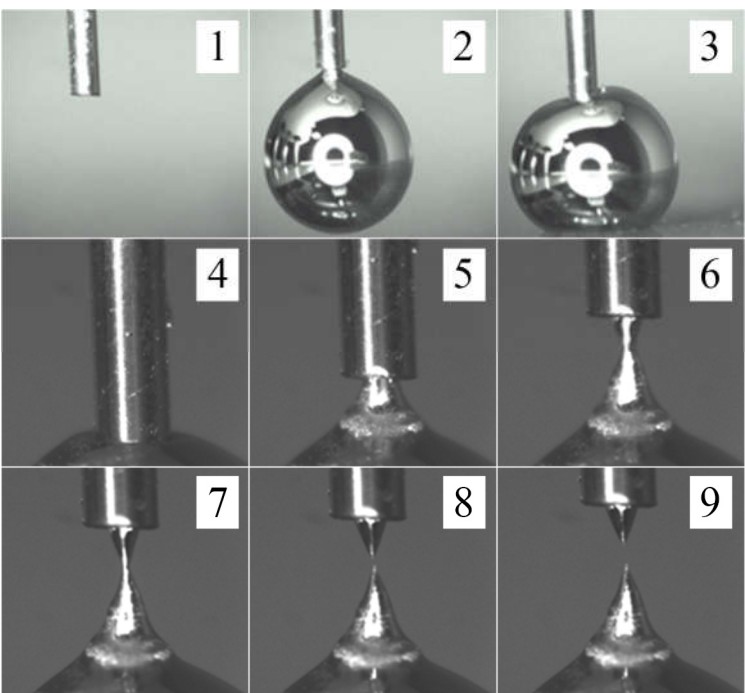

**Figure 1.** Formation of the EGaIn liquid eutectic electrode.

1. The conveying of the probe to the substrate, while adjusting the position of the optical microscope to observe the point of contact between the probe and the substrate;

2. The formation of a hanging drop suspended from the needle channel (it is necessary to move the stepper motor by X mm to squeeze out X μL);

3. Converging the hanging drop on the tip of the syringe to the substrate at a speed of 4000 μm/s;

4. Monitoring the contact between the hanging drop and the substrate. The speed of convergence of the probe with the hanging drop to the substrate is decreased to 400 μm/s.

The droplet experiences deformation and the contact between the droplet and the substrate increases. The convergence of the needle and the substrate is then stopped;

5. The beginning of the probe withdrawal from the substrate at a speed of 400 μm/s, showing the observable stretching of the meniscus between the drop and the needle channel;

6. The formation of a narrow transition from InGa in the probe to the drop on the substrate;

7. The formation of a cone-shaped probe. While observing the transition diameter of the order of 100 μm in an optical microscope, the withdrawal speed of the probe from the droplet is reduced to 40 μm/s;

8. Control of the shape of the formed probe. With a successful outcome, a drop of a well-defined conical shape will remain on the tip of the needle. In case of failure, when the drop takes the form of a sphere, the whole process should be started from step 2;

9. Retraction of the received probe from the sacrificial drop (by increasing the retraction speed). (Figure 1).

The next task is to change the position of the optical microscope to form the next probe. Following the formation of cone-shaped probes on the needles, the substrate on which traces remain after the formation of the InGa probes is replaced with a test sample. The optical microscope is then set up to observe the two probes in contact with the test sample.

As a result of this operation, an electrode is formed, as shown in Figure 2, which is an image taken with an optical microscope.

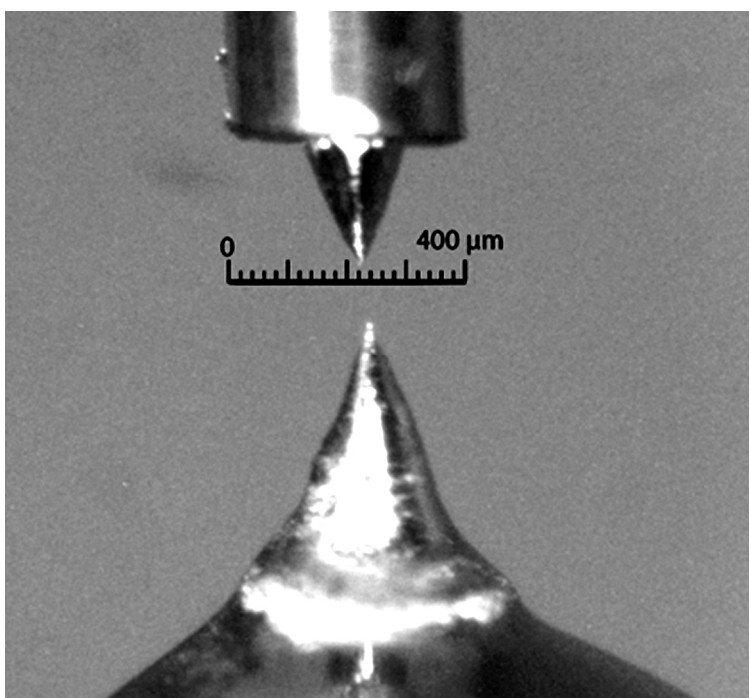

**Figure 2.** Photo of a formed conical eutectic liquid electrode.

As can be seen from Figure 2, this electrode has a conical shape, while its narrowest part reaches a diameter of about 10 μm (the diameter of the insulin syringe needle is 400 μm).

When choosing the speed of movement of the needle, it should be taken into account that at a low speed of movement, the formation of complex capillaries is possible, as in Figure 3.

By changing the distance between the liquid electrode and the measured sample, the contact surface can be changed (increased).

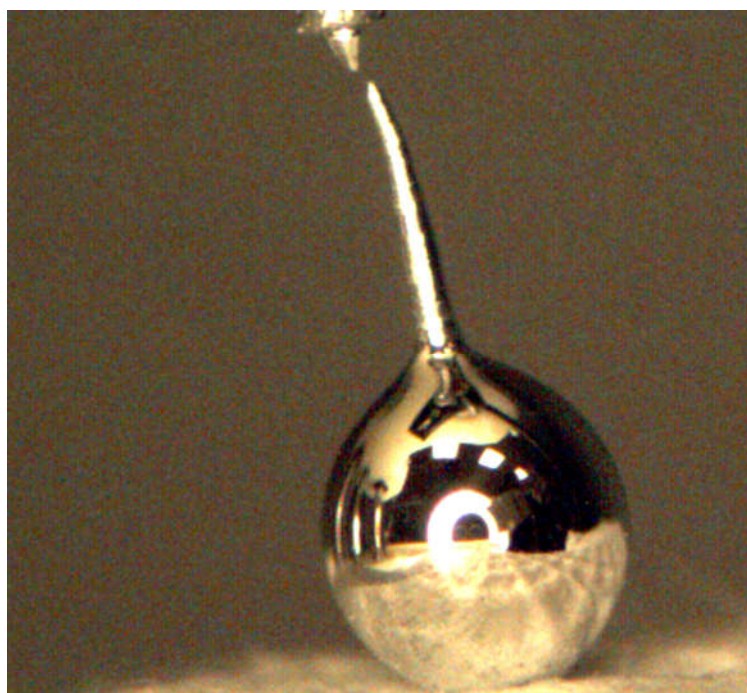

**Figure 3.** Extended outgrowth formed by the needle channel at a slow retraction rate.

### 2.2. Algorithm of the Operation with One Electrode

1. A certain amount of liquid eutectic EGain melt is collected into a syringe mounted on the movable part of the installation;

2. An electrical contact is placed on the needle of the syringe;

3. The operator, controlling the stepper motor of the installation from a personal computer, receives a cone-shaped eutectic electrode at the end of the needle, controlling the entire process on the computer screen using an optical microscope;

4. Once the liquid electrodes are obtained, the needle is retracted a to sufficient distance from the object table, and the sample is placed on the object table;

5. By controlling the position of the needle along the vertical axis with the help of an optical microscope on the computer screen, the liquid eutectic electrode is brought closer to the material under study until the moment they make contact. In this case, the contact of the electrode with the sample can be detected both visually, i.e., on the monitor of a personal computer, and by the current jump when a certain voltage is applied to the "sample-electrode" system;

6. Once the electrode is installed on the sample and there is a stable electrical contact, the current–voltage characteristic of the sample is measured. For this, a Tektronix PWS4323 digital programmable power supply and a Tektronix DMM4020 digital multimeter are used in conjunction with the LabView software package, which is used for digitizing and automating the measurements of the voltage–current characteristics;

7. For electrical measurements, a serial connection of a measuring multimeter and a power supply is used. In this case, the power supply operates in the voltage source mode, and the multimeter in the ammeter mode registers the current flowing through the circuit.

The installation was designed in the KOMPAS-3D v16 3D design program and then the parts were printed on a 3D printer. The final assembly of the unit is shown in Figure 4 (assembly was carried out using 6mm shafts, lm6uu bearings, 5mm studs and Kysan motors).

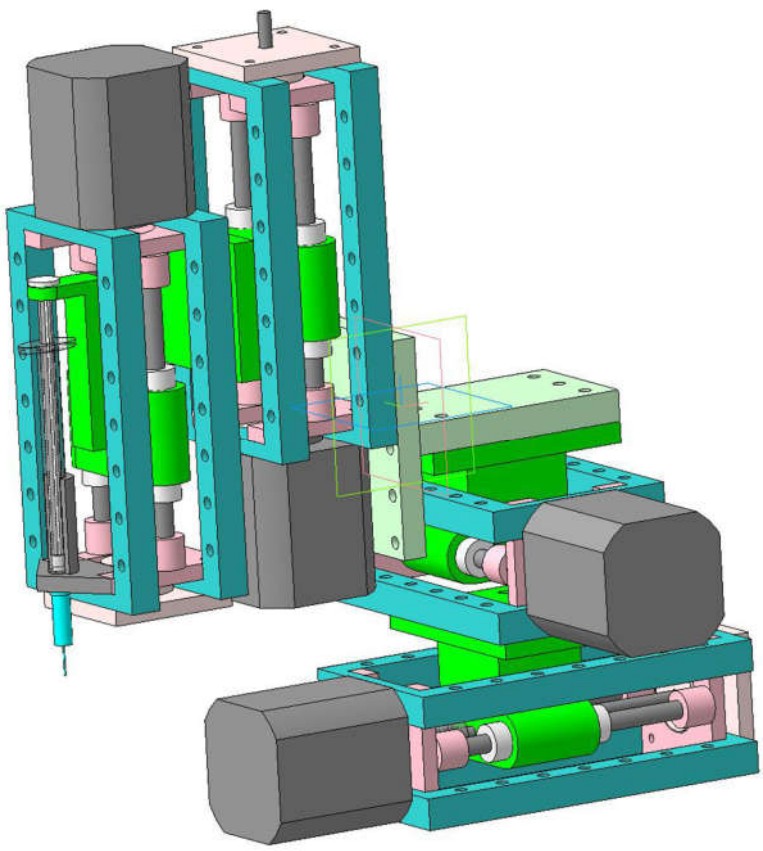

**Figure 4.** 3D assembly model for one probe.

## 3. Results and Discussion

According to scanning electron microscopy (SEM), the zinc oxide test samples have a developed morphology, as shown in Figure 5.

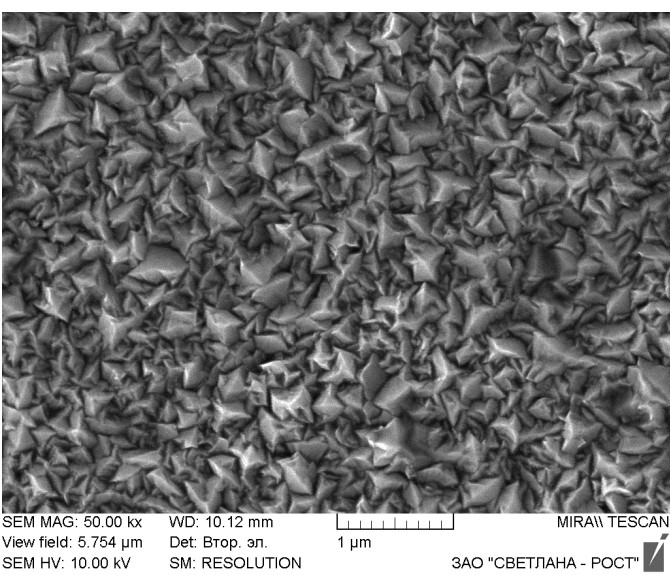

**Figure 5.** Example of an SEM image of the surface of a zinc oxide test sample.

For the measurements, an insulin syringe with a needle (Figure 6a) and a spring contact (P100-B1 brand) were used. The spring contact was fused into the plastic tip of the syringe (Figure 6b). The diameter of this probe is 500 μm.

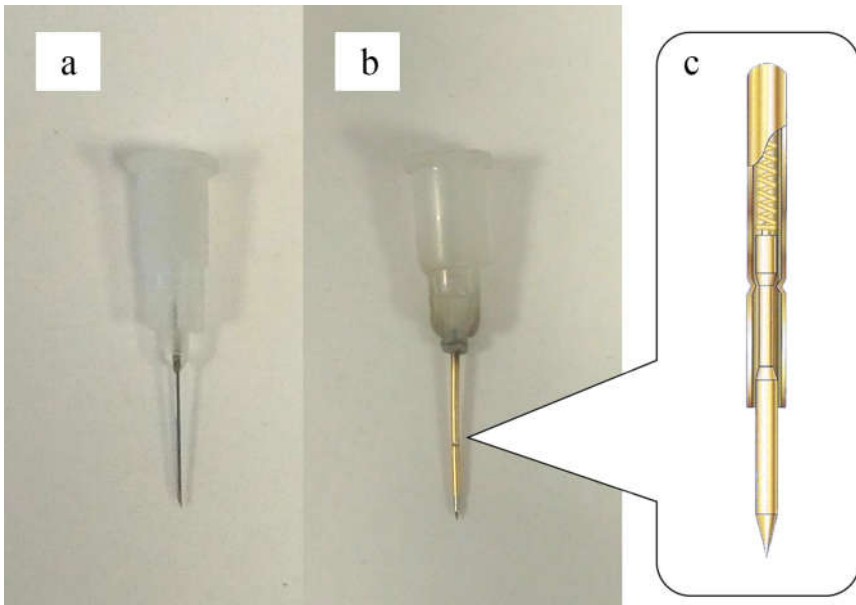

**Figure 6.** Modified rigid electrode: (**a**)—head of an insulin syringe with a needle; (**b**)—the head of an insulin syringe with a springy pressure contact fused into it; (**c**)—spring clamping contact.

A different contact area of the liquid eutectic electrode with the test sample can be obtained by changing the distance between the electrode and the sample. Various options for the contact area of the liquid electrode are shown in Figure 7, when studying a zinc oxide (ZnO) film. A—contact area diameter d = 40 μm; b—contact area diameter d = 80 μm; c—diameter of the contact area d = 120 μm.

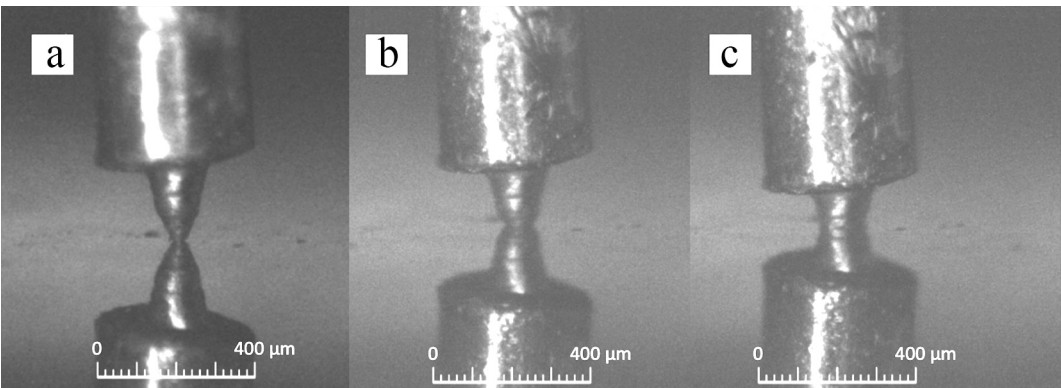

**Figure 7.** Various variants of the liquid electrode contact area: (**a**)—contact area diameter d = 40 μm; (**b**)—contact area diameter d = 80 μm; (**c**)—diameter of the contact area d = 120 μm.

Figures 8 and 9 show the current–voltage characteristics for three different electrodes corresponding to Figure 7a–c.

As can be seen from the dependencies obtained, at small contact areas (electrodes a (contact area diameter 40 μm) and b (contact area diameter 80 μm)), the current flowing through the sample is hardly distinguishable (tens of μA). However, at a large contact area (electrode c (contact area diameter 120 μm)), the current through the sample steadily increases sharply (up to 1 mA); this is clearly seen at a positive voltage.

As can be seen, there are regions of nonlinearity in the I-U characteristics. We assume that the oxide film forms folds upon contact with the surface of the sample under study. It turns out that the current at the "liquid electrode-sample" contact flows through a limited number of spots with a small thickness of the oxide layer.

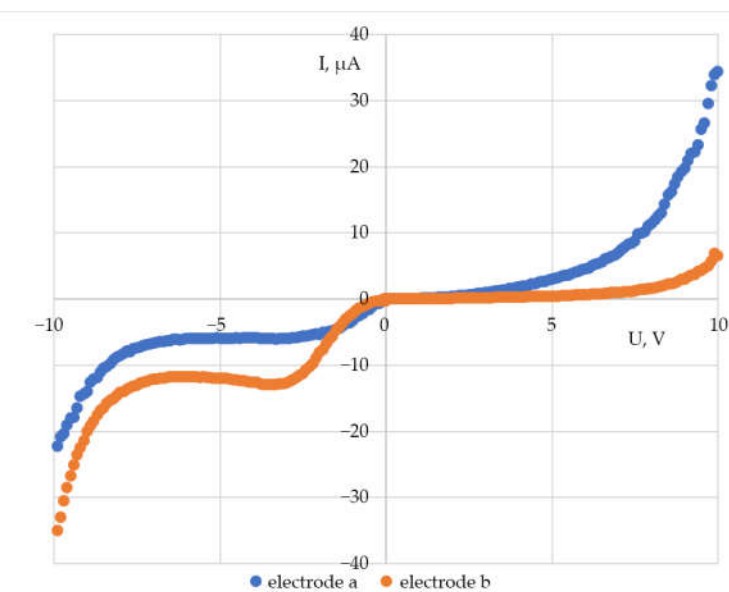

**Figure 8.** The current–voltage characteristics of the sample taken with electrodes (a) and (b).

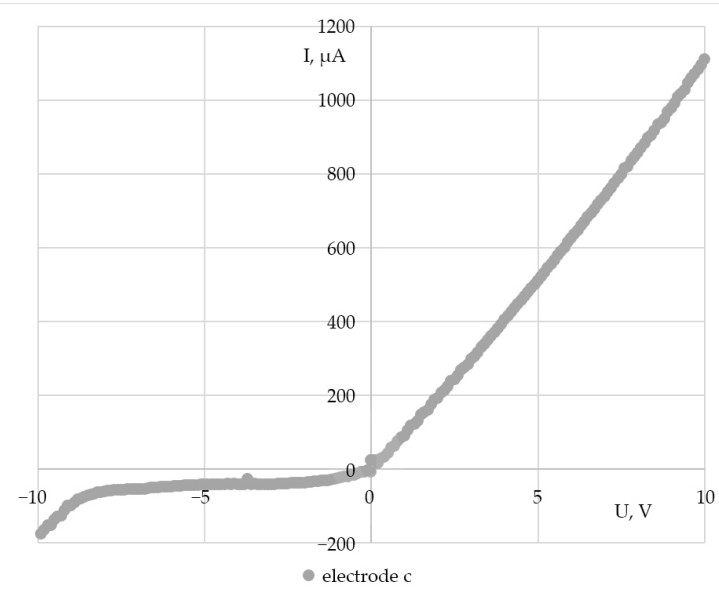

**Figure 9.** The current–voltage characteristics of the sample taken with electrode (c).

That is, the current distribution is not uniform in the cross section of the liquid electrode contact area, and this is more noticeable at small contact area sizes.

For the clamping probe, it is possible to vary the sample-probe holder distance, while due to the spring contact, it is possible to create a change in pressure on the sample.

The position of the spring head in each of the experiments is shown in Figure 10.

The type of current–voltage characteristics obtained is shown in Figure 11.

As can be seen from the dependencies obtained at high pressure, the conical tip of the electrode begins to deform the sample under study, thereby penetrating into the thin film (contact between a solid conical indenter and an elastic half-space, according to the Hertz model). As a result, the electrode is no longer on the surface of the thin film, but in its bulk, the contact area increases and the contact resistance decreases. At a high clamping pressure, the current through the sample is comparable to the current when using a large-area liquid contact. In this case, the presence of an oxide phase on the liquid electrode leads to the observation of a barrier on the current–voltage characteristic. The results of the comparison

of the current–voltage characteristics of a liquid eutectic electrode with a small contact area diameter and a rigid electrode with a small clamping pressure are shown in Figure 12.

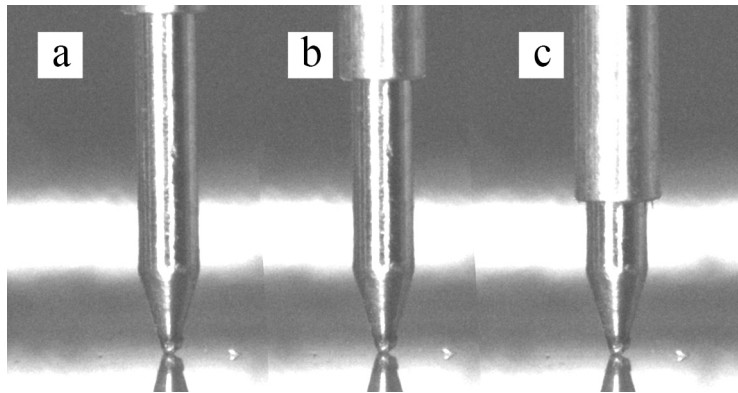

**Figure 10.** Rigid electrode positions exerting different clamping pressure on the surface Pa< Pb < Pc.

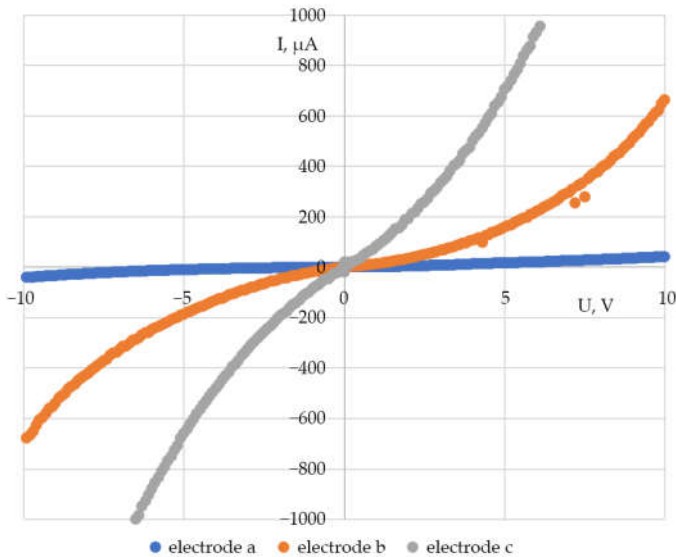

**Figure 11.** The current–voltage characteristics of the sample with three different electrode pressures on it: Pa < Pb < Pc.

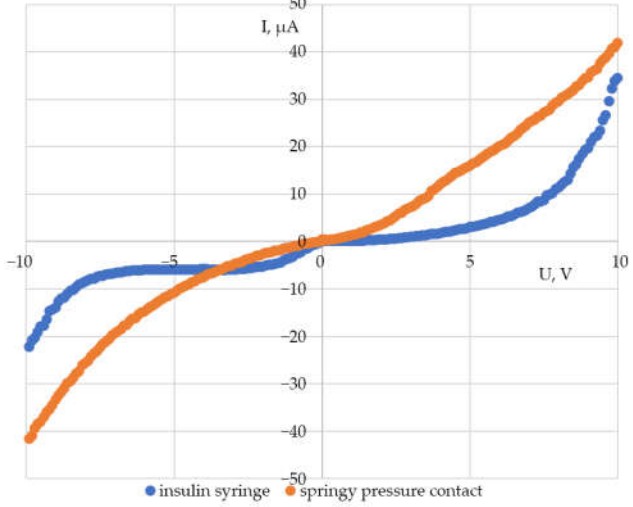

**Figure 12.** Comparison of the current–voltage characteristics of the sample obtained using a liquid and a rigid electrode.

As can be seen from the dependencies obtained, the current values do not differ much when using a liquid electrode with a small contact area and a rigid electrode with a low clamping pressure. However, for a rigid contact, the form of the obtained characteristic has a more linear character. These two measurement methods have their own peculiarities, taking into account the diameter of the contact area and the clamping force. Since the sample was not deformed when the current–voltage characteristic was taken using liquid contact, and the thin film did not experience pressure, the current–voltage characteristic obtained is more detailed for this sample, but the presence of the oxide phase will lead to the observation of the barrier.

## 4. Conclusions

Liquid metal eutectic alloys based on gallium–indium are currently widely used in various applications ranging from electronics to biomedicine. As part of the work, a setup was created that can be used to obtain contacts that do not mechanically deform a layer of material, as well as to explore the possibility of using conductive eutectic GaIn melts as electrodes for analyzing materials. The specific algorithms for obtaining a cone-shaped electrode and an algorithm of operation with one electrode, examples of obtained voltage–current characteristics of the test samples are proposed.

We assume that the observed regions of nonlinearity in the current–voltage characteristics are explained by the presence of an oxide film, which is always present on the surface of the EGaIn probe. The oxide film forms wrinkles upon contact with the surface. The current at the "drop-sample" contact flows through a limited number of conductive spots, that is, the current density is not uniform in the cross section of the contact spot.

The prerequisite for such a study was the desire to exclude pressure, and, as a consequence, deformation of the material surface that occurs when using solid clamping contacts. This problem is especially critical in the study of thin films, in which a solid electrode can scratch the upper part of the surface, thereby damaging the structure of the thin film.

**Author Contributions:** N.P.—paper writing, technology, methodology; E.M.—review of publications on the topic of the article, methodology, writing—review and editing; A.B.—technology, illustrations for the article; R.M.—technology, translation of the article into English, V.M.—conceptualization, data curation, interpretation of the obtained results, editing. All authors have read and agreed to the published version of the manuscript.

**Funding:** This research was funded by the Russian Science Foundation (grant no. 22-29-20162), https://rscf.ru/project/22-29-20162/ (accessed on February 2023) with the St. Petersburg Science Foundation (agreement No. 19/2022 dated 14 April 2022).

**Institutional Review Board Statement:** Not applicable.

**Informed Consent Statement:** Not applicable.

**Data Availability Statement:** Not applicable.

**Acknowledgments:** The authors are grateful to Pavel Somov (TESCAN) for SEM data.

**Conflicts of Interest:** The authors declare no conflict of interest.

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
