# Peer review of "Investigation of the Conductive Properties of ZnO Thin Films Using Liquid Probes and Creation of a Setup Using Liquid Probes EGaIn for Studing the Conductive Properties of Thin Films"

_technologies, doi:10.3390/technologies11010026_

Round 1
Reviewer 1 Report
The manuscript " Investigation of the conductive properties of ZnO thin films using liquid probes" written by R. Permiakov et al describes measuring the current–voltage characteristics of zinc oxide thin films at different contact areas of the liquid eutectic electrode with the test sample. There are a lot of questions to the authors about their results:
At first, the choice of the object for testing the methodology for measuring current-voltage characteristics is not clear. Why were chosen zinc oxide and such method for thin film preparation? As shown on Fig. 5, the surface of the sample is rough and it is formed from large microcrystalline particles. This can lead to poor or unreproducible contact between the metal eutectic droplet and the sample surface. What can be observed in Figures 8 and 9, where changes in the contact area lead to a significant scattering in the I-V data. Again, this technique does not provide a clean and contaminant-free surface, and any unwashed organic impurities can contribute to the conductivity. Since the authors claim a new technique for conducting current-voltage measurements, it would be better to take a sample with a smooth and clean surface and known characteristics in order to confirm the capabilities of their technique.
Secondly, there are still a number of questions about eutectics. As follows from introduction it consists from a liquid metal component that interacts with the gas atmosphere and a solid oxide layer, which can also undergo changes. How do contacts behave over time? Does the chemical composition and their mechanical characteristics change, does cracking occur. Does it depend on humidity, etc.?
Finally, as for the measurements themselves. The paper presents 8 current-voltage curves, and it is not clear which one to believe and the authors do not give a confident answer. To solve this problem, it is necessary to manufacture a measuring structure with reliable contact pads and measure the stable current-voltage characteristics of the synthesized zinc oxide films in order to determine which of the methods is correct and draw conclusions based on these new and reliable data.
Based on the foregoing, I can accept this article for publication only after major revisions.
Author Response
The authors are grateful to the Reviewer comments and useful suggestions to the manuscript. The responses are in the attached file.

Reviewer 2 Report
The language of the work is imprecise, it should be more concise. Drawing descriptions and comments are not always consistent. The work requires proofreading by a native speaker.
Line 115-138; needs linguistic correction
Fig1. maybe a better caption: Formation of EGaIn liquid eutectic electrode
Fig 2. maybe a better caption: By changing the distance between the liquid electrode and the sample, the contact surface can be changed.
similar in line 151
Line 190: relief? maybe it's better to use the word “morphology?”
Figs. 8,9, 11, 12-should be: voltage-current characteristics/characteristic- also in the text see for example line 221
Figs7, 8, 9 -explain the dependence of the current on the size of the contact surface - it is probably not a linear relationship?
Fig 11. Is it three different electrodes or three different pressures? Inconsistency of Fig.11 with the text under the drawing.
Fig 10. maybe a better caption: electrode positions exerting different pressures. You can't really see a change in the contact area - is it supposed to be like that?
The insert to Fig.11 is blurred
Line 226 stronger applied voltage?; stronger is applied electric field, the applied voltage is higher
Line 226-230: This is pure speculation or very naive. Obviously, as the voltage increases, the current also increases. With the increase of the pressure of the electrode to the sample, the contact surface also increases, which may cause an increase in the current
Line 243-250: This requires a deeper analysis. However, these characteristics are very different. For example, the blue characteristic shows clearly rectifying characteristics. What is the reason for this - please explain. However, it needs to be commented on more.
In principle, the presented conclusions/conclusion are based on Fig.12. Maybe it needs to be developed more, because I don't really agree with the thesis that these are almost the same characteristics.
Author Response

(The authors gave the same response as above.)

Reviewer 3 Report
Comments and Suggestions for Authors:
The manuscript entitled: „Investigation of the conductive properties of ZnO thin films using liquid probes” reports an attempt to apply liquid probes based on indium-gallium eutectic (EGaIn) for the study of current-voltage characteristics and charge transport in ZnO-based thin films.
The article creates a positive impression in general. It is clearly written and well organized, however, it would be good the authors address to some comments below before the publication of the submitted paper. Therefore, I suggest minor revision.
1. Introduction section has lots of basics. The authors should highlight the originality and the novelty of this work.
2. The inset in Fig. 11 is hardly visible. The axis captions in the inset are illegible. The figure must be improved to be clear for the readers.
3. The Conclusions section is too short. It should be supported by the results.
I believe the article will be acceptable for publication after rewriting the manuscript according to the aforementioned comments.
Author Response

(The authors gave the same response as above.)

Round 2
Reviewer 1 Report
In the revised version the authors improved the text significantly and now it can be published.
Reviewer 2 Report
Regardless of the situation in the world, science must progress, good luck!